# Ferdinand II of Aragon (1479–1516)

**Marta Serrano-Coll**

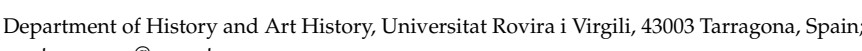

Department of History and Art History, Universitat Rovira i Virgili, 43003 Tarragona, Spain; marta.serrano@urv.cat

**Definition:** Ferdinand II king of Aragon (1479–1516). He was the fourth king of the Trastámara dynasty, which had first come to power after the Compromise of Caspe, reached after Martin I died with no living descendants in 1410. Although in terms of artistic patronage Ferdinand II was not as active as his wife Elisabeth I, he was still aware that the wise use of artistic commissions in reinforcing ideas and concepts favourable to the institution of the monarchy. He is a highly important figure in the history of Spain because, along with Elisabeth, he was one of the Catholic Monarchs and thus represents a new conception of power based on their joint governance, a fact that is reflected in the iconography found in his artistic commissions across all genres. All of the images are evidence of how King Ferdinand, at the end of the Middle Ages, wanted to be recognised by his subjects, who also used his image for legitimising and propagandistic purposes. Nobody else in the history of the Hispanic kingdoms had their image represented so many times and on such diverse occasions as did the Catholic Monarchs.

**Keywords:** royal images; royal iconography; kings of Aragon; crown of Aragon; Fernando II of Aragon

---

## 1. Introduction to the Reign of Ferdinand II

Ferdinand II was not destined to be king, he was born after the second marriage of Johan II of Aragon (1458–1479) to Juana Enriquez, and was the king's second son. The crown should have gone to Charles, Prince of Viana and son of Blanche of Navarra. However, the clashes and hostilities convulsing the kingdom meant that the Aragonese Cortes of 1461 decided that the second son should succeed to the throne. The climate remained convulsive until the death of Johan II, when Ferdinand was unanimously accepted. All of his subjects, including the Catalans, pinned their hopes on him.

On 5 March 1469 Elisabeth, who had been proclaimed heir to the crown of Castile in the Treaty of Toros de Guisando, signed the Capitulations of Cervera, which meant she entered into a marriage agreement with the heir of Aragon, Ferdinand. Together and as equals their reign was to be one of the most important in the history of Spain and would mark the future of the peninsular kingdoms. Ferdinand's concern for the defence of Christianity was internationally recognised; he was commemorated as "Ferdinand, the Catholic King, propagator of the Christian empire", in the inscription accompanying his wreathed portrayal in the Vatican *stanzas* painted by the famous Rafael.

Under the Catholic Monarchs Spanish national unity was still *de facto* rather than *de jure*; nevertheless, their reign was central to the history of Spain and the creation of the modern nation (on just the subject of his kingdom, see [1–6]). The death of Ferdinand II ushered in a new era in the history of the kingdom of Aragon with the accession of Charles I of Spain and V of Germany, a member of the Habsburg dynasty who assumed the government of Castile, Navarre and Aragon and came to personify one of the most powerful kingdoms in modern times.

## 2. Character, Appearance and Artistic Patronage

We do not have in-depth knowledge of the king's character and appearance despite the information provided by chroniclers and travellers who alluded to him. Perhaps Hernando del Pulgar's physical description is the most accurate: "he was a man of medium height, well-proportioned in his limbs, in the features of his well-composed face, his eyes smiling, his hair tight and smooth [...]. His speech was even, neither hurried nor too slow. He was of good understanding and very temperate in eating and drinking, and in the movements of his person [...] neither anger nor pleasure altered him [...]. He was a great hunter of birds, and a man of good effort and a hard worker in war [...]. And he had a singular grace that anyone who spoke with him immediately esteemed him and wished to serve him [...]" [7].

He was seduced by pieces of jewellery, especially if they had diamonds and rubies. Some of these pieces were made by famous silversmiths, the records showing that there were as many as eight in his service, one of whom was Jewish [8]. He enjoyed showing off his jewellery and on one occasion he even survived an attack in Barcelona on 7 December 1492 because the width of his necklace prevented the knife of his would-be assassin, Joan de Canyamàs, from penetrating deep enough to kill him. The episode was recorded in the margin of two pages of the *Dietari del Consell de la Ciutat de Barcelona* (Arxiu Històric de la Ciutat, Barcelona. Ms. A-359), perhaps by the scribe Marc Bosquets, who details the event and the punishment suffered by the attacker [9] (authorship proposed by [10]; analysis of drawings in [11]). It is surprising to learn that he was illiterate, although as a Renaissance prince he did much to promote culture, as did his wife Elisabeth. It is said that he inspired Machiavelli's work "The Prince" (among others, [12]).

Both Ferdinand and Elisabeth exploited the royal image and increased its prestige through court ceremonials, panegyrics, and iconography, for which they used novel, rich and varied artistic forms which were open to Renaissance trends, although without excluding the late Gothic, Islamic and *Mudejar* styles, which persisted in architecture, objects and everyday settings. Their image proliferated in various media, accompanied by extensive inscriptions, heraldry, and the use of devices such as the yoke and arrows to allude to the names of the monarchs, and the Gordian knot, related to the motto of *tanto monta* that summed up the equality between them as heads of government. Ferdinand II was aware that art was the most visible sign of his power and he always commissioned works in conjunction with his wife, to the extent that once he was widowed, he continued with the works they had planned or begun. He should be considered one of the great patrons of the Hispanic Middle Ages, and although he was served by artists of lesser status than those who worked for his wife, one can still find renowned names such as the painters Tomás Giner, Miguel Ximénez and Hernando del Rincón, the silversmith Jaume Aymerich, the miniaturist Alonso Ximènez and the sculptors Gil Morlanes and Domenico Fancelli (the following studies by Joaquín Yarza are essential reading [13–17]).

## 3. Elements of a Legal Nature: Coins and Seals

### 3.1. Coins

Fernando II continued with the previous coins types, although he also opened a new period that led to new types and iconography. The result of the new artistic experiences was the integration of his portrait into his dies, something unusual in the numismatic trajectory of the kings of Aragon.

Continuing the policy of his predecessors, he unified the values of the traditional coins in all his territories. He generalised the use of the *ducat* or *ducat d'or*, also called the *excelente* in the Valencian mint [18] (Figure 1). With a diversity of dies according to their denominations and places of issue, the hitherto consecrated profile of bust/shield contrasts with the introduction of the new typology F or F and Y crowned/shield and, above all, with the original representation of the busts facing each other/shield.

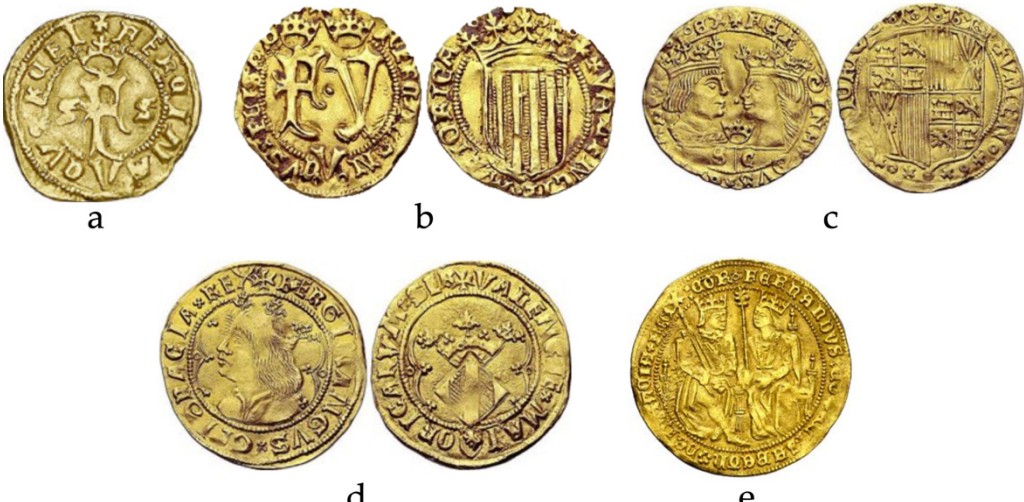

**Figure 1.** Coins of Fernando I. (**a**). *Ducat* of Valencia, with F, obverse; (**b**). *Ducat* of Valencia, with F and Y crowned, obverse and reverse; (**c**). *Doble ducat* or *Excelente* of Valencia, obverse and reverse. All from https://www.numisbids.com/n.php?p=sale&sid=359&cid=10127 (accessed on 20 October 2021); (**d**). *Doble ducat*, obverse and reverse. From https://www.numismaticodigital.com/noticia/5525/ultima-hora/hoy-seleccion-500-de-aureo&calico-en-barcelona.html (accessed on 20 October 2021); (**e**). *Doble castellano* or *dineral*, obverse. From https://aureocalico.bidinside.com/es/lot/2010/reyes-catlicos-sevilla-doble-castellano-/ (accessed on 20 October 2021).

The crowned initials, perhaps originating from the miniature [19], had precedents in Castile and Leon (see variants in [20]), although they can also be seen in the coinage of Johan II, father of Ferdinand, king consort of Navarra (in his *blancas* and *medias blancas* of made of copper and silver alloy. The Prince of Viana also minted *gruesos* with his crowned initial. See [21]). The iconography of the images facing each other: "with the face of us and of the most honourable queen our wife", ordered by Ferdinand in his commission to García Gomis, regent of the mint of Valencia in 1488 [22], also had more immediate precedents in Castile. It arose from the reform generated by the Ordinance of 1475, which established this gold coin and stipulated that it had to display the frontal busts of the kings, their names, and the titles of their kingdoms, while silver coins were introduced featuring the coat of arms of the yoke and arrows and the aforementioned crowned initials. For the first time, both monarchs were depicted together on the coinage of Seville, thus reflecting the new governmental model (on the monetary reforms of 1475 and 1497, which confirm the concept of two-headed government, see [23,24]). After Elisabeth's death in 1504, this coin underwent modifications; the effigy of the queen on the obverse and the arms of Castile and Leon on the reverse would disappear. The new coins would advertise Ferdinand's new status, with the Castilians referring to him disparagingly as *catalanote* and insisting that he was only king of Aragon. They would feature the traditional bust of the king on the obverse and a crowned lozenge with the arms of Aragon on the reverse [25]. It was a brief minting; with the death of Philip I, Ferdinand II regained control of Castile, meaning that his coins also returned to their previous imagery.

The *doble castellano* or *dineral*, which features the enthroned sovereigns on the obverse, was a new introduction in the Iberian Peninsula. Its iconography had been established in the Royal Decree of 1475 [26–28] and was new in the Hispanic territories. Undoubtedly, the collecting ancient coins and medals by the high dignitaries of the court led to knowledge of this typology, typical of Byzantine coinage until the 13th century, and which also reflected the political reality of the joint-government established by the two monarchs (details on the iconography on the coinage of Ferdinand II, also outside the peninsular kingdoms, see [11], pp. 19–32).

### 3.2. Seals

Ferdinand II continued to use certain earlier typologies, as is evidenced by his main seals, which are almost identical to those of John II except for details and legends [29]. Leaving aside his minor seals, all of which are heraldic, his bulls are particularly interesting, these being two types of metallic stamp of varying dimensions. The first is the traditional one: equestrian/heraldic, although on the reverse the Saracen heads are face-on and crowned. The second has a new feature: the obverse depicts the equestrian sovereign and the reverse the enthroned queen (Figure 2).

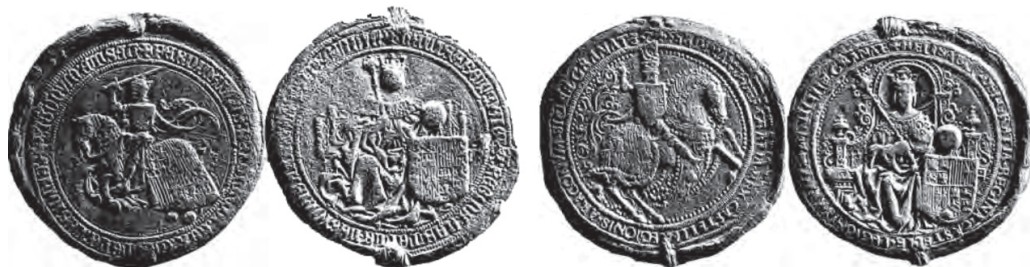

**Figure 2.** Lead Bulls of the Catholic Monarchs. Undated. Published by [29], nums. 112, 131.

On the obverse, surrounded by + FERDINANDVS: DEI: GRACIA: REX: CASTELE: LEGIONIS : ARAGONVM : ET SEC, we can see the king mounted on his horse, which is facing either right or left and appears less light of foot than its predecessors because its protective coverings are more rigid. Perhaps this is because of the need to incorporate the complex arms of the Catholics Monarchs and would also explain why the rider's shield is unemblazoned. On the reverse, encircled by + HELISABET: DEI GRA: REGINA: CASTELLE: LEGIONIS ARAGONVM: ET SECILIE, the queen is enthroned and accompanied by a shield displaying an emblem identical to that of the rider's coat of arms. There are numerous pieces, and with slight variations; some of them betray elements of the new trends in monumental sculpture at the time, referred to by some as *plateresco* because of its connections with works in precious metals.

Although they invert the iconographic order (equestrian/ enthroned), the traditionalism of these pieces, in accordance with the models of the Crown of Aragon, should not deceive: these bulls represent the first appearance of the royal couple on the same seal, thus providing a visual depiction, as seen on their coins, of their joint governance.

## 4. Instrumental Character of Art

### 4.1. Government Images

It is striking to note the virtual absence of any images of Ferdinand showing him exercising his ministry, *in sede maiestatis*, a pose so common among his predecessors. During his reign, emblems became so prominent that they pervaded coins and seals, and came to occupy the place of the effigies of the sovereign who, alone or in the company of notaries, scribes or members of the court, in initials or in separate vignettes, attested or validated the document they headed. The transposition of numismatic and sigillographic models to miniatures continued to be common, as is illustrated on fol. 2r of the *Privilegios de la Santa Cruz de Valladolid*, from 1484 (preserved in the Biblioteca de la Universidad, Valladolid, doc. 9), which derives from the *excelentes* or *medio excelentes* (see [14], p. 454 and [11], pp. 43–44), to cite one example.

### 4.2. The King as Caput Milicie

King Ferdinand was the object of adulation by patrons, private individuals or members of secular and religious institutions. This can be seen, for example, in the most outstanding artistic project undertaken by Cardinal Mendoza, namely, the lower stalls of Toledo Cathedral. In this work, the cardinal exalted Ferdinand and Elisabeth in a remarkable manner (Figure 3) by also extolling himself for his close collaboration with them in the war

against Granada. Chiseled by Rodrigo Aleman between 1489–1495, it was begun before the conclusion of the campaign, which demonstrates its patron's conviction that this holy war would have a successful outcome [30–32]. The fact that the cardinal is depicted seven times, six times with the king and once with both monarchs, is evidence of the benefit to be gained from appearing in effigy alongside the Catholic Monarchs (see, [11], p. 56).

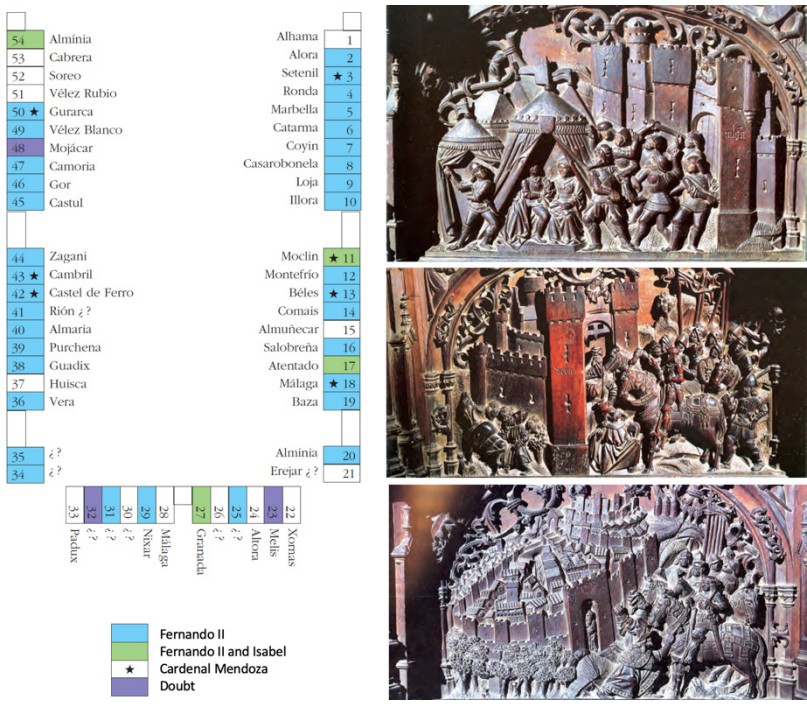

**Figure 3.** Diagram of the *sillería* with its protagonists. 1489–1495. Detail of the stalls: 17. Attempt against the Monarchs in Malaga; 36. Surrender of Vera; 27. Handing over the keys of Granada. Published by [30].

Having become analogous with the Reconquista as noted Müntzer (according to [30], p. 16), Ferdinand and Elisabeth are depicted in triumphal scenes, mostly showing city authorities surrendering and handing over their keys, or the entry of the sovereign into subjugated towns, although sometimes other anecdotal episodes are sculpted, which the sculptor may have learnt about as the war progressed. The presence of this military chronicle in a cathedral setting can be explained by the fact that the war with Granada was not only a political act but was also a crusade blessed by God [33] (see, also, [14], p. 456 and [11], pp. 54–93).

*4.3. Devotional Images*

During the reign of Ferdinand II, the use of devotional objects as vehicles for political propaganda continued. Although there are precedents, the use of iconography as a pretext or structure under which complex symbolic programmes were concealed became systematised and generalised.

Exemplary in this respect are the Plasencia stalls by master craftsman Rodrigo Aleman, who was contracted by the representatives of the cathedral chapter on 7 June 1497 (Figure 4). The two chairs at the ends of the stalls, together with the central one for St Peter, are the largest and stand on a special base that gives their occupants a commanding view and, at the same time, allows them to be easily seen (see [33], p. 104 and [34]). Both present inlays of the Catholic Monarchs, who had the prerogative of accessing the choir as honorary canons and collecting the corresponding *ratione* -prebend or benefice-, a custom that spread in the late Middle Ages probably due to the more direct intervention of kings in ecclesiastical affairs (see [35,36]. The chairs' dimensions and position on high, similar to that of the

venerable Peter, place the monarchs in a glorious spatial environment, a new visual sign of their supposed sacredness that the monarchs so longed for (see [14], p. 467).

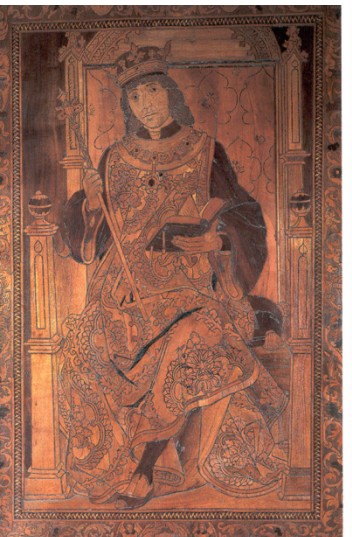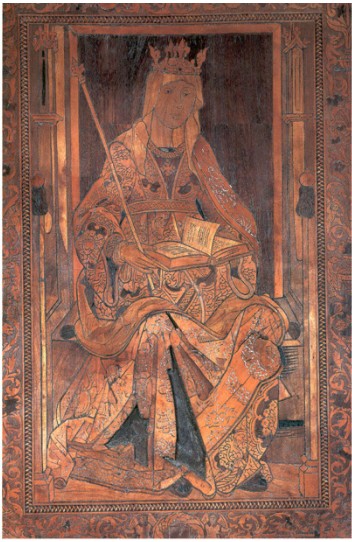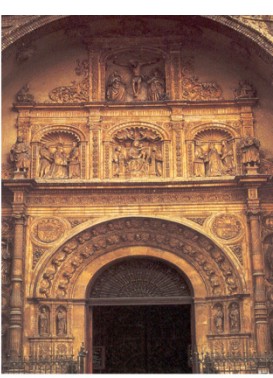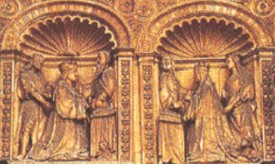

**Figure 4.** The Catholic Monarchs in the Plasencia cathedral stalls. 1497–1503. Published by [35] vol. II, p. 138; Santa Engracia monastery. 1514–1516. General view and detail of the Reyes Católicos. Published by [8], p. 239.

The monarchy's desire to make its presence felt in the religious sphere was manifested in other developments, as is illustrated by the portals of the monastery of Santa Cruz in Segovia, the church of El Paular, the most problematic portal of the cloister of Segovia Cathedral (descriptions and problems in [11], pp. 118–124), and the well-known portal of Santa Engracia in Saragossa (Figure 4). The latter was commenced by Ferdinand II's father, Johan II, who, after entrusting himself to the saint, had his sight restored after a cataract operation in 1468 [37–39]. When John II realized that he would not be able to complete it, he commissioned Ferdinand to do so, given that he "liked to see the designs, because he had a taste for architecture" [40]. To this end, Ferdinand II wrote, on 8 May 1493, that "the work on the Aljafería should cease and everything that was to be spent there should be redirected to the work on Santa Engracia" [41]. Catalogued as one of the earliest examples of a Renaissance doorway in Spain, and executed by the Morlanes family, its iconography features several elements, including the monarchs, the ancient cults of the sanctuary, symbols of the order that took over the monastery, and the connotations underlying the form and ornamentation of the triumphal arch that constituted the doorway. It was a showcase of intentions at a time when the king sought to dignify his image, which had deteriorated in Catalonia due to the civil war against his father, and in Castile, where his power was questioned by the nobility (see [8], p. 64). Some believe the effigy of the king is a portrait, either because of a sculpture that was kept in the sacristy of the monastery or because Gil Morlanes the Elder maintained a close personal relationship with the monarchs [42] (see, also [37], p. 13).

The images depicting the king as protector and restorer of the Church, and as an exemplary and just devotee, mostly together with his wife, are very common. This can be seen in the doorway of the collegiate church of Daroca, which dates to around 1482–1488, proof of his predilection for important sanctuaries, in this case dedicated to the *Sagrados Corporales*, to which he allocated resources for their restoration and embellishment [43,44] (see, also, [8], p. 79). Another example is the anonymous *Piedad de los Reyes Católicos* in the cathedral of Granada, perhaps an ex-voto donated by the monarchs on their second entry into the city on 5 January 1492 [45], or the *Mater Omnium* of Santa María la Real de Las Huelgas, from around 1485 by Diego de la Cruz and his workshop, the result of the imposition of Leonor Mendoza as abbess, despite the opposition of the community. In a

context of tension, the abbess or her uncle, the famous cardinal, endowed the monastery with a work that showed the union within the community and its links with the royalty, who had extended such favours towards it [46] (and [14], p. 465).

Other religiously and politically significant representations are those that allude to religious orthodoxy and spiritual renewal. One of the most illustrative works is the famous panel of the *Virgen de los Reyes Católicos* of Saint Thomas of Avila (nowadays in Museo del Prado, Madrid), from around 1490 and closely related to the Holy Inquisition [47] (Figure 5). The institution was lauded by the monarchy because, in addition to looking after the interests of the Church, it enabled the monarchs to wield unquestioned power in each of their kingdoms (see [4], pp. 134–135). The attention to detail and the coincidence with the descriptions of these monarchs leads us to think that their portraits were painted in their presence or from sketches of them taken during their lives [48] (see, also, [42], p. 51). What is certain is that this panel is an indication that the Inquisition had royal and divine approval [49]: not only do the two patron saints of the convent appear, but alongside the kings are two other Dominican inquisitors, Pedro de Arbués, martyred in Saragossa by opponents of the Inquisition, and Tomás de Torquemada, who was prior of the monastery (according to [8], pp. 35–38; [48], planche LVIII and [50]). This panel, an early court portrait that is predominantly devotional in character, is propaganda in defence of the Court of the Holy Office, a fact that is corroborated by the presence of its most prominent members (one of whom was martyred for its cause) and of the sovereigns (who worked so hard for its reinstatement).

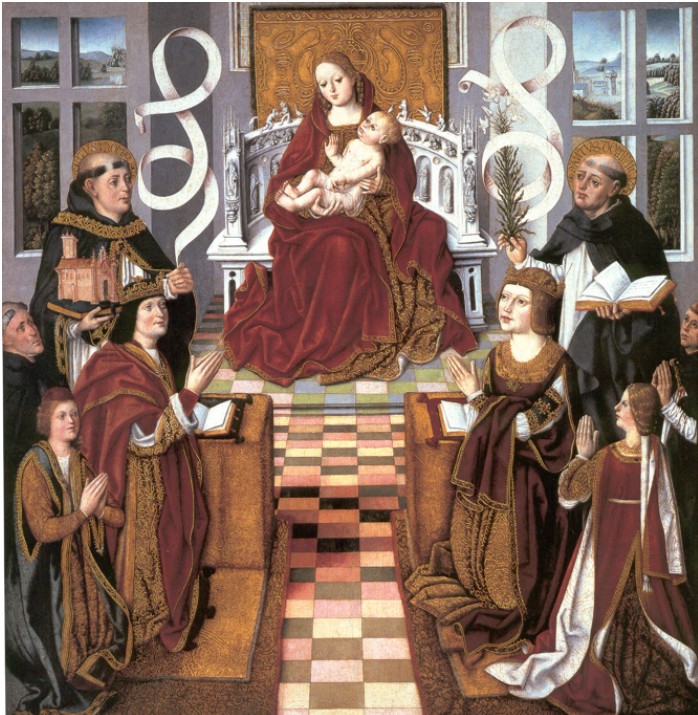

**Figure 5.** *Virgen de los Reyes Católicos*. c. 1490. Published by Bango, I. Dir.; *Maravillas*, vol. II, p. 184.

## 5. A New Artistic Genre at Court: Portraiture

Portraiture was introduced at court in the time of the Catholic Monarchs. In addition to the aforementioned early portraits in the *Virgen de los Reyes*, the *Mater Omnium* and, in sculpture, on the façade of Santa Engracia, there were other examples, such as the portrayals that appeared in some scenes of the *Políptico de Isabel la Católica* (this set contained 47 little panels), of which 28 panels have survived, two with effigies of Ferdinand II. Perhaps his painter, John of Flanders, used this work as a pretext to paint the kings from life [51,52].

This genre reflected, in image and likeness, the true portrait of the king [53]. The institutional framework in which the monarch wanted to be seen, with the insignia of his

status, was no longer important; instead he wanted a faithful record of his appearance. Earlier attempts had been made: John I (1387–1396) in 1388 tried to hire Jacques Coene after learning of his skills in depicting particular faces [54]. Ferdinand II also lamented his attempt to secure the hand in marriage of the Neapolitan *Infanta* for his son John, which failed because he lacked a painter of sufficient quality to be able to send a suitable likeness of him (see [14], p. 444 and [55]).

The new genre was intended to be a mirror and record of individual features. There are 4 known examples of King Ferdinand II, practically identical and following the compositional formula of the Flemish portrait in the 15th century: the Windsor portrait, from around 1490–1500; the Vienna portrait, of the same date; the Berlin portrait, after 1492; and the Poitiers portrait, of the same date [56,57] (see, also, [11], chap. VI (Figure 6).

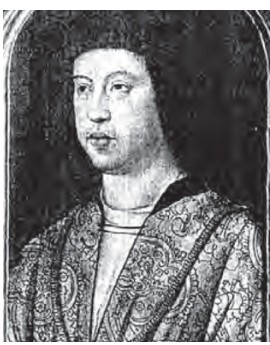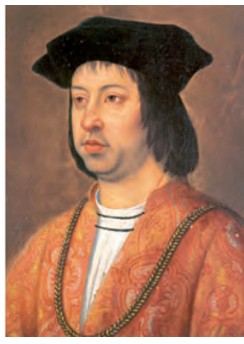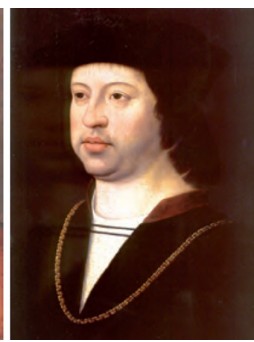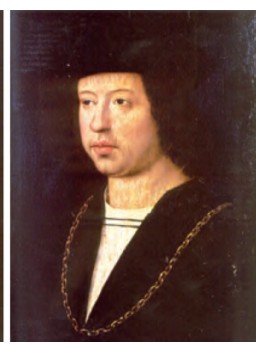

**Figure 6.** Fernando II portraits: Palacio Real, Windsor Castle. c. 1490–1500. Published by [55], planche VI; Kunsthistorishes Museum, Viena. c. 1490–1500. Published by Schütz, K.; Vitale, A. Anonimo fiammingo. Rittrato di Ferdinando II di Aragona, detto il Cattolico. In: *I Borgia. L'arte del potere*. Electa: Roma, Italy, 2002, p. 10; Preussischer Kulturbesitz, Berlin. Post. 1492. Published by *Reyes y mecenas*, p. 375; Museum of Poitiers. Published by Fernández, *Fernando*, p. 373.

The greatest similarities are to be found between the Windsor and Vienna portraits (the other two being simpler), the differences being limited to the colour of the clothes and the necklaces on his chest. These similarities suggest that they were not painted from life; moreover, the precision of the details and features of the king's adult face indicate that portraiture as an independent genre had become fully established in the Iberian Peninsula, an art form hitherto almost unknown in Spain.

## 6. Conclusions

Ferdinand II is one of the great personalities related to the image of the king of Aragon. Firstly, a new conception of power based on joint government with Elisabeth was witnessed and reflected in the iconography in all artistic genres, with the most representative media being seals and coins, stamped at their behest and whose surfaces shared, for the first time, the effigies of both kings. Secondly, the Catholic Monarchs were the object of adulation on the part of the artistic patrons among their subjects, whether these were private individuals or members of secular or religious institutions, and they personified the exaltation of the monarchy to a hitherto unseen extent, although always in keeping with the clear instrumental nature of the artistic projects, including those promoted by the monarchs themselves. Regarded as *caput milicie* and true defenders of the faith, which earned them the nickname of the Catholic Monarchs, they continued the already established use of sacred works as true vehicles of political propaganda, and under their rule the use of iconography as a pretext or structure for concealing complex symbolic ideas became systematic and generalized.

**Funding:** This research was funded by *Edificis i Escenaris religiosos medievals a la Corona d'Aragó*, [2017 SGR 1724]. Generalitat de Catalunya-AGAUR.

**Conflicts of Interest:** The author declares no conflict of interest.

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
