# Peer review of "Ferdinand II of Aragon (1479–1516)"

_encyclopedia, doi:10.3390/encyclopedia1040089_

Round 1

Reviewer 1 Report

I highly rate this article. It is brief, substantive in content, and well written. The text is logical, and the arguments are convincing. Historical facts are well presented. Chosen illustrations correspond perfectly with the main text. Also, the bibliography is rich and helpful for the reader. 

Author Response

Thank you really much for your comments, and thank you for your time.

Reviewer 2 Report

We suggest to the author some proposals, in order to clarify some data for a reader who does not know the historiography of the Crown of Aragon.

line 18: Juan II of Aragon (1458-1479)

line 24: it is suggested to incorporate a sentence, at the beginning, explaining his marriage to Isabella of Castile. Then continue with “Together with his wife Isabel...”

line 54: it is suggested to add a sentence explaining the relationship of King Ferdinand with the literary work of Machiavelli's Prince.

line 61: it is suggested to incorporate a photograph with the joint heraldry of Kings Isabella and Ferdinand.  

For example in an illuminated manuscript owned by both kings: The Morgan Library, MS M.1044, fols. 1v: https://www.themorgan.org/collection/livre-de-la-chasse/2

line 61: You may also consider adding one or two lines specifying the mottoes and symbolic images: yoke and arrows. It is also suggested to add a picture.

line 96-97: I propose the substitution of "Philip the Handsome" for Philip I of Castile.

line 99: Figure 1- I don't know if there is any difficulty with the photographs of the Ducat de Valencia, with F and Y crowned. They appear to be a little blurry. If they can't be seen well on screen I suggest you consider deleting them.

lines 319-320: note 20. Specify the volume of this publication: volume I.

Author Response

First of all, thank you very much for the attention with which you have read my work. I will take all your considerations into account.
Once again, thank you very much for your comments, which have undoubtedly improved the result of my work

Reviewer 3 Report

My suggestions appear in the attached document.

Author Response

(The authors gave the same response as above.)

Round 2

Reviewer 2 Report

No changes are proposed

Reviewer 3 Report

No more comments. I've already gave some to the past version.